# Changes in visual acuity and retinal microstructures following vitrectomy for lamellar macular hole

Hitoshi Goto[1], Noriko Kubota[2]*, Yosai Mori[3], Kazunori Miyata[3], Yuji Nakano[2], Tomoyuki Kunishige[2], Fumiki Okamoto[2]

1 Department of Ophthalmology, Nippon Medical School Tama-Nagayama Hospital, Tokyo, Japan,
2 Department of Ophthalmology, Nippon Medical School, Tokyo, Japan, 3 Miyata Eye Hospital, Miyazaki, Japan

* oishinoriko@nms.ac.jp

## Abstract

This study aimed to evaluate the changes in visual outcomes and optical coherence tomography (OCT) findings in patients with lamellar macular hole (LMH) following vitrectomy. Consecutive patients diagnosed with LMH based on OCT findings who underwent vitrectomy between April 2020 and December 2023 were included. Forty-two patients (male, n = 21; female, n = 21; mean age, 71.3 ± 8.2 years) were included in the study. Preoperative and postoperative best-corrected visual acuity (BCVA) and OCT parameters, including the presence of inner and outer retinal cysts, epiretinal proliferation (EP), and ellipsoid zone disruption (EZ), were analyzed. Postoperative BCVA was assessed at 6 months after surgery, and postoperative OCT findings were evaluated at 1, 3, and 6 months using all available data. Mean BCVA significantly improved from 0.36 ± 0.33 logarithm of the minimal angle of resolution preoperatively to 0.15 ± 0.26 at 6 months postoperatively (p < 0.001). The frequency of eyes with inner retinal cysts was 21.4% preoperatively, which gradually decreased to 8.0% at 6 months postoperatively. The frequency of outer retinal cysts was 54.8% preoperatively, which significantly decreased to 12.0% at 6 months postoperatively (p = 0.00937). The presence of EP was significantly correlated with worse postoperative BCVA (p < 0.05). Both preoperative and postoperative EZ disruptions significantly correlated with worse postoperative BCVA (p < 0.05). Vitrectomy for LMH improves visual acuity and reduces most intraretinal cysts within 6 months. EP, EZ disruption, and poor preoperative visual acuity were identified as factors associated with poor visual outcomes in patients with LMH.

## Introduction

Lamellar macular hole (LMH) is a retinal condition characterized by a partial-thickness defect in the foveal area, often leading to visual disturbances such as

**Data availability statement:** The data underlying this study contain potentially identifying or sensitive patient information and cannot be made publicly available due to ethical restrictions imposed by the Central Ethics Committee of Nippon Medical School Foundation (Approval No. M-2023-161). Qualified researchers may request access to the data by contacting the Central Ethics Committee of Nippon Medical School Foundation (email: chuorinri.group@nms.ac.jp, Tokyo, Japan).

**Funding:** The author(s) received no specific funding for this work.

**Competing interests:** The authors have declared that no competing interests exist.

metamorphopsia and reduced visual acuity [1–3]. Recently, Hubschman et al. [4] established a consensus definition of optical coherence tomography (OCT) characteristics and clarified the diagnoses of LMH, macular pseudohole (MPH), and epiretinal membrane–foveoschisis (ERM-FS), previously known as tractional LMH.

Vitrectomy is a widely used surgical approach for LMH that aims to restore retinal structure and improve visual outcomes [3,5–7]. However, because the pathogenesis of LMH does not mainly involve tangential retinal traction [2], it is difficult to expect any effect from the removal of the ERM or internal limiting membrane (ILM). Recently, a surgical technique was devised in which an epiretinal proliferation (EP), composed mainly of Müller cells, is embedded into a foveal tissue defect, and its usefulness has been reported [7–12]. However, various studies have discussed the management of EP and ILM, with differing conclusions [3,8,9,13]. Furthermore, the prognostic factors influencing surgical outcomes remain an area of active investigation. Preoperative best-corrected visual acuity (BCVA) and the presence of ellipsoid zone (EZ) disruption have been proposed as the key predictors of postoperative success [6].

To the best of our knowledge, there is currently no consensus on the optimal indications for surgery for LMH, and follow-up data are limited. This study aimed to evaluate the visual outcomes and prognostic factors in patients undergoing vitrectomy for LMH. Analyzing both anatomical and functional changes would help in determining the efficacy of surgical intervention and the predictors of postoperative visual improvement.

## Materials and methods

This retrospective study was conducted at Nippon Medical School, Japan, and was approved by the Ethics Committee of Nippon Medical School [14] (ID: M-2023–161). All investigative procedures adhered to the principles of the Declaration of Helsinki. Written informed consent was obtained from all patients for their information to be stored in the hospital database and used for research purposes. Data for this study were accessed between 2/10/2024 and 30/11/2024. All data were fully anonymized before analysis.

### Patient data and OCT analysis

Forty-two eyes from 42 patients who underwent pars plana vitrectomy (PPV) between April 2020 and December 2023 were reviewed. All patients underwent examinations using spectral-domain OCT (Spectralis version 1.8.6.0; Heidelberg Engineering GmbH, Heidelberg, Germany). Both horizontal and vertical OCT B-scan images were obtained. LMH was diagnosed based on OCT criteria, including an abnormal foveal contour, a foveal cavity with overhanging edges, and indicators of foveal tissue loss, as defined by the recent consensus [4]. ERM was defined as hyperreflective proliferation on the surface of the ILM, and the associated conditions, such as MPH and ERM-FS, were also classified based on updated criteria proposed by the recent consensus [4]. Eyes meeting more than two diagnostic criteria for different associated diseases across OCT B-scan images were considered overlapping cases and included in the analysis of each corresponding group. Patients with LMH associated

with other retinal diseases such as diabetic maculopathy, retinal vein occlusion, age-related macular degeneration, and active uveitis were excluded. The analyzed OCT parameters included the presence of inner and outer retinal cysts, EP, and EZ disruption. Inner and outer retinal cysts were defined as hyporeflective, small, round or elliptical spaces located in the inner nuclear layer and outer nuclear layer, respectively. EP was defined as homogeneous isoreflective epiretinal material over the ILM, and EZ disruption was defined as discontinuity of the EZ line. Fig 1 shows the serial OCT images of a representative case.

These parameters were evaluated preoperatively and at 1 month, 3 months, and 6 months postoperatively. To assess the diagnostic reproducibility, two examiners (H.G. and N.K.) independently evaluated all images. A consensus was reached between the investigators for all the eyes analyzed.

## Visual function assessment

Preoperative and postoperative BCVAs were measured using a standard Japanese decimal visual acuity chart at a distance of 5 m. For statistical analyses, decimal values were converted to the logarithm of the minimal angle of resolution (logMAR) units. Spherical equivalent (SE) and axial length (AL) were measured using an autorefractometer (RC-5000; Tomey Corporation, Nagoya, Japan) and ultrasonography (AL-4000; Tomey Corporation), respectively.

## Surgical procedures

Vitrectomy was performed using a 25-gauge system by two experienced surgeons (Y.M. and F.O.). After removing the lens via phacoemulsification, an intraocular lens was implanted when required, followed by vitrectomy. In all cases,

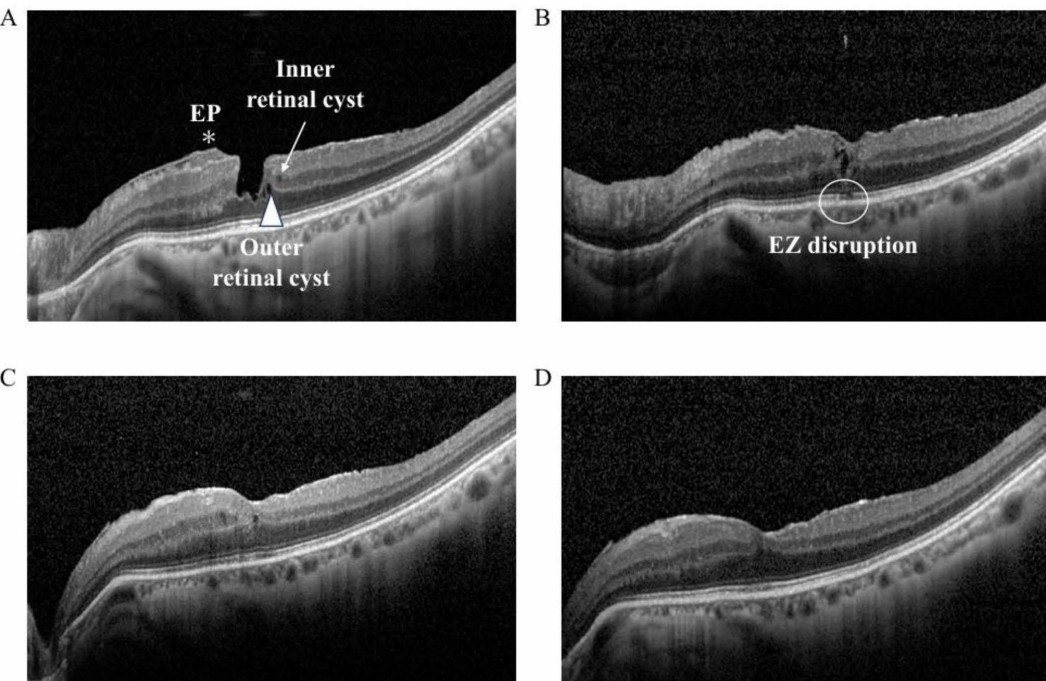

**Fig 1. A representative serial OCT image of inner and outer retinal cysts, epiretinal proliferation, and ellipsoid zone disruption in an eye with lamellar macular hole (LMH).** The arrow indicates inner retinal cysts, and the arrowhead highlights outer retinal cysts. The asterisk denotes epiretinal proliferation (EP), and the circle marks ellipsoid zone (EZ) disruption. (A) Baseline: EP, inner and outer retinal cysts are observed. (B) One month postoperatively: EP has resolved, but both inner and outer retinal cysts remain. The EZ appears indistinct. (C) Three months postoperatively: Outer retinal cysts have resolved, while inner retinal cysts persist. The EZ appears continuous. (D) Six months postoperatively: Inner retinal cysts have also resolved.

0.1–0.2 mL of 0.025% brilliant blue G solution was injected gently over the macula and then washed out using an irrigation solution. After the ERM was peeled, an additional 0.1 mL of brilliant blue G solution was applied to stain the ILM. The ILM was peeled off. In eyes with EP, an inverted ILM flap was embedded into the foveal defect. Tamponade was performed using balanced salt solution (BSS) or air.

## Statistical analyses

Mean scores and standard deviations were calculated for age, AL, SE, and preoperative and postoperative BCVAs. The difference between the preoperative and postoperative BCVAs was analyzed using the Wilcoxon signed-rank test. Spearman's rank correlation test was used to examine the correlation between preoperative and postoperative BCVAs. The relationships between preoperative and postoperative OCT findings and postoperative BCVA were analyzed using the Mann–Whitney U test. Analysis of the categorical data in cross tables, such as the presence of inner and outer retinal cysts, EP, and EZ disruption, was performed using McNemar's chi-squared tests. For the longitudinal comparison of categorical OCT parameters, three pre-specified paired comparisons (Pre vs 1M, Pre vs 3M, and Pre vs 6M) were evaluated, and Bonferroni correction was applied (adjusted significance threshold: $p < 0.0167$). All other tests of association were considered statistically significant at $p < 0.05$. The analyses were performed using SPSS Statistics version 29.0 (IBM Corp., Armonk, NY, USA).

## Results

In total, 42 eyes from 42 patients (male, n = 21; female, n = 21; mean age, 71.3 ± 8.2 years) were analyzed. Among all eyes, 35 (83.3%) were classified as having mixed LMH subtypes, including 14 eyes (33.3%) with ERM-FS, 18 eyes (42.9%) with MPH, and 3 eyes (7.1%) with both subtypes. Table 1 shows the demographic and clinical characteristics of the patients with LMH. Thirty-eight of the 42 eyes underwent vitrectomy combined with cataract surgery. Tamponade was performed using BSS in 12 eyes and air in 30 eyes. No significant intraoperative complications were observed. The number of eyes available for OCT analysis at each follow-up time point was as follows: 42 eyes preoperatively, 40 at 1 month, 38 at 3 months, and 25 at 6 months. Six eyes with glaucoma were also included. All of these had early-stage disease, with MD values better than −3 dB and no visual field loss within the central 5 degrees. Eyes with more advanced glaucoma that could affect central visual acuity were not included in the analysis.

BCVA significantly improved from 0.36 ± 0.33 preoperatively to 0.15 ± 0.26 at 6 months postoperatively (p < 0.001). A significant correlation was observed between the preoperative and postoperative BCVAs (p < 0.001, r = 0.522; Fig 2). EP

**Table 1. Demographics and clinical characteristics of patients with LMH.**

| Number of eyes | | 42 |
|---|---|---|
| Mixed type | + ERM-FS | 14 (33.3%) |
| | + MPH | 18 (42.9%) |
| | + MPH+ERM-FS | 3 (7.1%) |
| Sex (Men/Women) | | 21/ 21 |
| Age (years) | | 71.3±8.2 |
| Prevalence of glaucoma | | 6 (15.8%) |
| Axial length (mm) | | 24.9±1.88 |
| Preoperative spherical equivalent (diopter) | | −2.59±4.57 |
| Preoperative BCVA (logMAR) | | 0.36±0.33 |

Values are presented as the mean ± standard deviation.

ERM = epiretinal membrane, MPH = macular pseudohole, ERM-FS = epiretinal membrane-foveoschisis, BCVA = best corrected visual acuity.

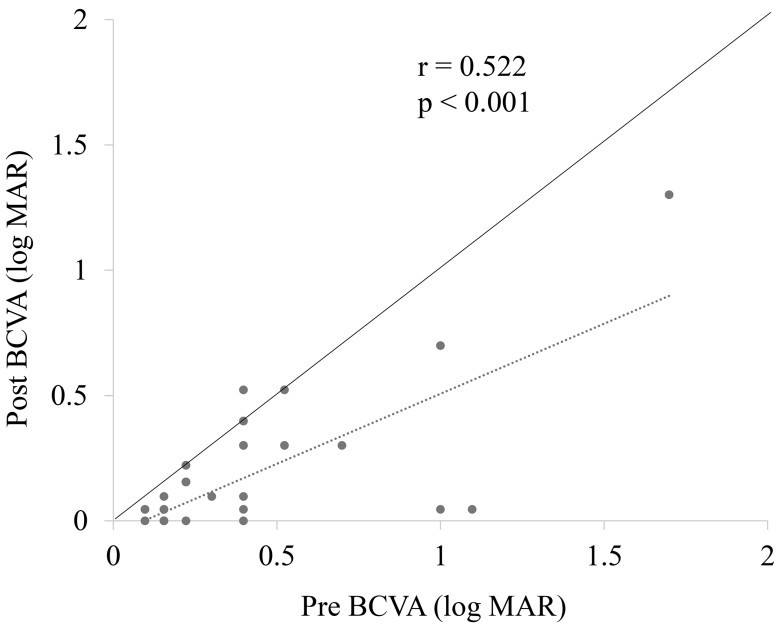

**Fig 2. Correlation between preoperative and postoperative best-corrected visual acuity (BCVA) in patients with lamellar macular hole (LMH).** A significant correlation was observed between preoperative and postoperative BCVA (p < 0.001, r = 0.522).

was observed preoperatively in 59.5% of the eyes. Fig 3 shows the frequency of inner retinal cysts, outer retinal cysts, and EZ disruption in the eyes preoperatively and at 1 month, 3 months, and 6 months postoperatively. The frequency of inner retinal cysts was 21.4% preoperatively and gradually decreased thereafter. At 6 months, the frequency decreased to 8.0%, but the change was not statistically significant. The outer retinal cysts were present in 54.8% of the cases before surgery, which decreased to 12.0% at 6 months, showing a statistically significant reduction. The frequency of eyes with EZ disruption decreased modestly postoperatively but did not change significantly during the 6-month follow-up period.

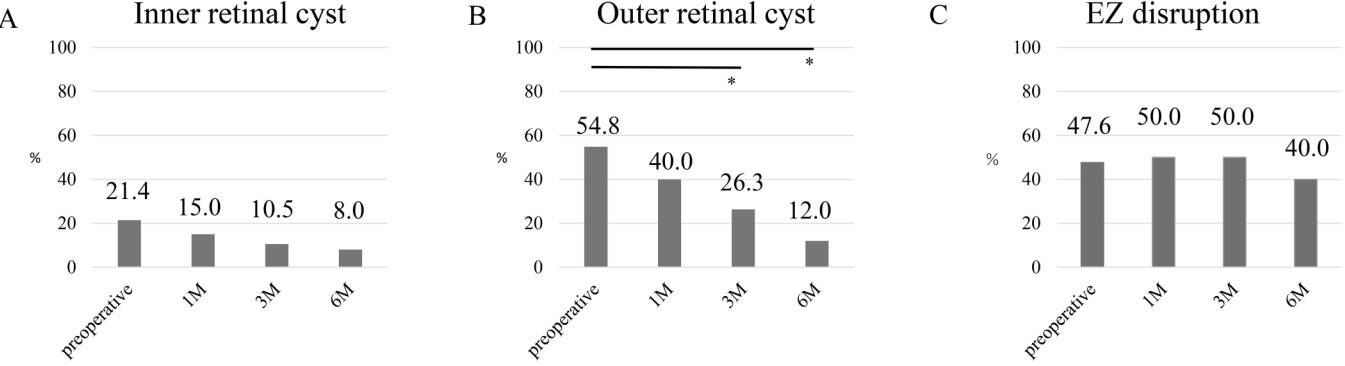

**Fig 3. Presence of inner retinal cyst, outer retinal cyst, and ellipsoid zone (EZ) disruption at each time point.** The frequency of eyes with inner retinal cysts (A), outer retinal cysts (B), and EZ disruption (C) at each time point. The frequency of inner retinal cysts was 21.4% before surgery and gradually declined after surgery. At 6 months, the frequency had decreased to 8.0%, although the change was not statistically significant. The outer retinal cysts were present in 54.8% of cases before surgery and decreased to 12.0% at 6 months, showing a statistically significant reduction. The frequency of eyes with EZ disruption did not change significantly during the 6-month follow-up period. McNemar's test with Bonferroni correction was applied. Significant differences were observed only for outer retinal cysts between preoperative and 3 months (p = 0.00328) and between preoperative and 6 months (p = 0.00937). * p < 0.0167 (Bonferroni-corrected).

In the subgroup analysis comparing eyes that received air tamponade (n = 30) with those that received BSS tamponade (n = 12), no significant differences in postoperative BCVA or OCT findings were observed at any follow-up time point.

Additionally, we performed a multiple linear regression analysis to identify factors associated with the 6-month postoperative BCVA, including preoperative BCVA and OCT parameters. The adjusted $R^2$ was 0.35, and the variance inflation factor for all variables was below 1.31, indicating no significant multicollinearity. Among the predictors, worse preoperative BCVA was significantly associated with worse postoperative BCVA (p = 0.0052).

Table 2 shows the association between the preoperative and postoperative OCT parameters and BCVA. Preoperative and postoperative BCVAs showed no association with inner and outer retinal cysts. Contrastingly, eyes with EZ disruption at baseline, 1 month, 3 months, and 6 months postoperatively exhibited worse postoperative BCVA. The presence of EP was significantly correlated with worse postoperative BCVA.

## Discussion

The clinical and OCT findings were investigated in patients with LMH, and the relationship between these factors and postoperative visual outcomes was analyzed in the present study. These findings provide insights into prognostic factors and highlight the importance of specific OCT parameters in guiding surgical management.

**Table 2. Relationships between pre- and postoperative OCT parameters and visual acuity in lamellar macular hole.**

| | | | Pre-BCVA | p-value | Post-BCVA | p-value |
|---|---|---|---|---|---|---|
| **Presence of inner retinal cyst** | Preoperative | (+) n = 9 | 0.43 ± 0.16 | 0.429 | 0.36 ± 0.28 | 0.962 |
| | | (-) n = 33 | 0.33 ± 0.32 | | 0.14 ± 0.26 | |
| | Month 1 | (+) n = 6 | 0.32 ± 0.15 | 0.745 | 0.21 ± 0.25 | 0.444 |
| | | (-) n = 34 | 0.37 ± 0.14 | | 0.36 ± 0.28 | |
| | Month 3 | (+) n = 4 | 0.40 ± 0.09 | 0.148 | 0.30 ± 0.27 | 0.166 |
| | | (-) n = 34 | 0.35 ± 0.36 | | 0.13 ± 0.28 | |
| | Month 6 | (+) n = 2 | 0.34 ± 0.26 | 0.8 | 0.26 ± 0.37 | 0.723 |
| | | (-) n = 23 | 0.37 ± 0.40 | | 0.15 ± 0.30 | |
| **Presence of outer retinal cyst** | Preoperative | (+) n = 23 | 0.41 ± 0.41 | 0.98 | 0.19 ± 0.33 | 0.989 |
| | | (-) n = 19 | 0.30 ± 0.17 | | 0.11 ± 0.15 | |
| | Month 1 | (+) n = 16 | 0.51 ± 0.45 | 0.114 | 0.22 ± 0.37 | 0.625 |
| | | (-) n = 24 | 0.27 ± 0.12 | | 0.10 ± 0.16 | |
| | Month 3 | (+) n = 10 | 0.53 ± 0.48 | 0.09 | 0.33 ± 0.43 | 0.128 |
| | | (-) n = 28 | 0.30 ± 0.25 | | 0.08 ± 0.16 | |
| | Month 6 | (+) n = 3 | 0.34 ± 0.22 | 0.832 | 0.19 ± 0.29 | 0.899 |
| | | (-) n = 22 | 0.37 ± 0.41 | | 0.16 ± 0.31 | |
| **EZ disruption** | Preoperative | (+) n = 20 | 0.42 ± 0.36 | 0.157 | 0.27 ± 0.37 | < 0.05* |
| | | (-) n = 22 | 0.28 ± 0.24 | | 0.03 ± 0.11 | |
| | Month 1 | (+) n = 20 | 0.40 ± 0.40 | 0.661 | 0.25 ± 0.38 | < 0.05* |
| | | (-) n = 20 | 0.32 ± 0.25 | | 0.04 ± 0.11 | |
| | Month 3 | (+) n = 19 | 0.41 ± 0.41 | 0.636 | 0.26 ± 0.39 | < 0.05* |
| | | (-) n = 19 | 0.30 ± 0.24 | | 0.04 ± 0.13 | |
| | Month 6 | (+) n = 10 | 0.44 ± 0.52 | 0.593 | 0.39 ± 0.44 | < 0.05* |
| | | (-) n = 15 | 0.31 ± 0.29 | | 0.05 ± 0.13 | |
| **Presence of EP** | Preoperative | (+) n = 25 | 0.41 ± 0.35 | 0.146 | 0.22 ± 0.30 | < 0.05* |
| | | (-) n = 17 | 0.26 ± 0.27 | | 0.02 ± 0.09 | |

*p < 0.05.
BCVA=best corrected visual acuity, EZ=ellipsoid zone.

Most patients with LMHs (83.3%) had a subtype of a related disease. A previous study revealed that 34.1% were of the mixed type, exhibiting at least two overlapping pathophysiologies of LMH, ERM-FS, and MPH [15]. Furthermore, a report indicated that these related diseases may transition into one another [16]. Based on these findings, achieving an accurate diagnosis of these related diseases requires the incorporation of OCT images from multiple directions and careful monitoring of temporal changes.

These results showed that vitrectomy significantly improved visual acuity and that postoperative BCVA was significantly correlated with preoperative BCVA. In a number of cases, the LMH remains stable over time, with no changes in its morphology or visual function. However, in some cases, central tissue defects expand while the EP slowly increases in size, and visual acuity tends to worsen; therefore, careful follow-up is necessary. Furthermore, some cases have developed full-thickness macular holes during follow-up, resulting in a significant decrease in visual acuity [3]. Figueroa et al. [5] reported that visual acuity improves 6 months after vitreous surgery in LMH (formerly "degenerative LMH") [17]. Similarly, Omoto et al. [6] reported that visual acuity did not improve 3 months after vitrectomy but improved over the long term. They also reported that preoperative visual acuity was related to postoperative visual acuity [6]. In this study, visual acuity improved significantly 6 months after vitrectomy. However, the better the preoperative visual acuity, the better the postoperative visual acuity, and these results were consistent with a previous report [6]. Giansanti F et al. reported that visual acuity continues to improve from 1 to 6 months after surgery and even later, indicating a slow and sustained retinal healing process [18]. Based on these findings, it is essential to avoid missing the appropriate timing for surgery and to inform patients that postoperative visual improvement may require some time. In addition, because preoperative BCVA appears to be an important indicator of postoperative visual outcomes, it may serve as a useful reference when determining the timing of surgical intervention.

Whether all LMH develop because of tractional forces acting on the fovea or because of inner retinal layer degeneration remains uncertain. Partial or complete posterior vitreous detachment, causing foveal traction, can potentially lead to the formation of intraretinal cysts/schisis or ERM. LMH may also be a consequence of the aborted full-thickness macular hole formation process [3]. Conversely, some cases of LMH do not show clear indications of traction. Instead, they are more closely associated with EP, composed of hyalocytes, fibroblasts, and glial cells that lack contractile properties, along with degeneration of the inner layers of the fovea, suggesting a distinct subtype of LMH [19]. However, recent research has demonstrated that both retinal traction and tissue loss play a role not only in LMH but also in related conditions, such as ERM-FS and MPH [15]. In vitrectomy for LMH, there is limited research on how changes in intraretinal cysts (inner and outer retinal cysts) affect visual function, making findings from related diseases highly valuable for reference. Inner retinal cysts have been reported to be associated with Müller cell damage [20]. Chen et al. [21] indicated that the presence of inner retinal cysts was associated with poorer postoperative visual outcomes in patients with ERMs. In contrast, most inner retinal cysts resolve postoperatively in patients with MPH [22]. The effect of inner retinal cysts on postoperative visual function remains controversial, with conflicting reports in the literature [22,23]. The frequency of inner retinal cysts showed a decreasing trend at 6 months postoperatively, but the difference was not statistically significant in this study. Furthermore, no significant correlation was observed between the presence of inner retinal cysts and preoperative or postoperative BCVA. Conversely, outer retinal cysts are thought to be caused by leakage from capillaries because of mechanical damage caused by retinal traction [24]. This hypothesis is supported by a report [25] indicating that the resolution of retinal traction through vitrectomy alone leads to improvement of outer retinal cysts associated with ERM and subsequent gains in visual acuity. In this study, the frequency of eyes with outer retinal cysts significantly decreased at 6 months postoperatively; however, no significant correlation was observed between the presence of outer retinal cysts and preoperative or postoperative BCVA. A considerable number of LMH cases overlapped with ERM and ERM-FS in these results. This overlap may account for the slight differences in the correlation between preoperative and postoperative BCVAs and the presence or absence of intraretinal and extraretinal cysts compared with previous reports of similar diseases. In this study, cystic changes alone appeared to have limited value in guiding surgical decision-making for LMH and were better interpreted together with other structural findings.

Greater emphasis should be placed on EZ disruption when evaluating its association with visual function in LMH. The association between EZ line integrity and visual function has been well-documented in various retinal diseases [26–28]. Nakamura, et al. [1] reported that the residual EZ index within the central 1-mm region was significantly correlated with BCVA in patients with MPH. The EZ, which contains a high density of photoreceptor cell mitochondria, plays a crucial role in photoreceptor cell function [29]. Damage to the EZ is thought to reflect photoreceptor cell dysfunction and is a factor in the loss of visual acuity. The frequency of eyes with EZ disruption did not change significantly during the 6-month follow-up, and eyes with EZ disruption at 3 and 6 months postoperatively exhibited worse postoperative BCVA despite no significant differences in preoperative BCVA. These findings suggest that eyes with preserved EZ integrity are more likely to benefit from surgery, whereas substantial EZ disruption may limit postoperative improvement, making EZ status a useful clinical indicator when considering surgical timing and expected outcomes.

Previous studies [1,10,30–32] have reported a high prevalence of EP in LMH, ranging from 36% to 100%. Yang, et al. [31] suggested that EP originated from Müller cells based on immunohistochemical analysis, and given that LMH is characterized by tissue loss at the fovea, EP may develop as a compensatory response to this loss. Specifically, a prior report [33] indicated that patients without EP achieve better postoperative visual outcomes than those with EP, which is consistent with the association observed in our study. However, this relationship was no longer evident after adjusting for preoperative BCVA, suggesting that EP may reflect baseline retinal status rather than serving as an independent prognostic factor.

EP-embedding surgery is effective for LMH with an EP [7–12]. Recently, EP-sparing surgery has been reported to be effective for treating LMH [13]. ILM peeling was performed, and an inverted ILM flap was embedded into the fovea in this study. These findings suggest that the surgical approach for LMH may need to be adjusted depending on the presence of EP. Matoba et al. [15] reported that, in the absence of EP, removal of ERM (and ILM) is suggested, and when EP is present, a combined approach of ERM (and ILM) removal and EP embedding (or sparing) can be considered. In the future, establishing embedding (or sparing) surgery for EP as a standard surgical treatment will require evidence of long-term outcomes.

This study has several limitations. First, the postoperative follow-up period was relatively short (6 months), and some cases were lost to follow-up, which may have reduced the statistical power of subsequent analyses. Furthermore, although EZ disruption and EP were associated with poorer postoperative BCVA, these associations were not observed in the multivariate analysis after adjusting for preoperative BCVA. This may suggest that their apparent effects reflect baseline retinal status rather than independent prognostic determinants. A larger sample size will be necessary to clarify their true influence. Second, the classification was based solely on OCT B-scan images, potentially limiting the accuracy of disease differentiation. Third, no direct comparison was made between overlapping and isolated cases, which may have affected the generalizability of the findings. Finally, there was no standardized surgical indication or uniformity in surgical techniques, which could have introduced variability in the outcomes.

## Conclusion

Visual acuity significantly improved after vitrectomy for LMH. Considerable overlap with related conditions was observed, and the presence of EP, EZ disruption, and poor preoperative visual acuity were identified as significant predictors of poor visual outcomes.

## Supporting information

**S1 Table. Comparison of pre- and postoperative visual acuity between eyes with and without air tamponade.**
(PDF)

**S2 Table. Comparison of pre- and postoperative OCT findings between eyes with and without air tamponade.**
(PDF)

## Author contributions

**Conceptualization:** Fumiki Okamoto.

**Data curation:** Hitoshi Goto, Noriko Kubota, Yuji Nakano, Tomoyuki Kunishige.

**Formal analysis:** Hitoshi Goto, Noriko Kubota.

**Investigation:** Yosai Mori, Kazunori Miyata.

**Project administration:** Fumiki Okamoto.

**Resources:** Yosai Mori, Kazunori Miyata.

**Supervision:** Fumiki Okamoto.

**Writing – original draft:** Hitoshi Goto.

**Writing – review & editing:** Hitoshi Goto, Noriko Kubota, Fumiki Okamoto.

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
