## [Decision Letter · Decision Letter 0]

28 Nov 2025

Dear Dr. Kubota,

Thank you for submitting your manuscript to PLOS ONE. After careful consideration, we feel that it has merit but does not fully meet PLOS ONE’s publication criteria as it currently stands. Therefore, we invite you to submit a revised version of the manuscript that addresses the points raised during the review process.

We look forward to receiving your revised manuscript.

Kind regards,

Tatsuya Inoue

Academic Editor

PLOS ONE

Journal Requirements:

Reviewers' comments:

Reviewer's Responses to Questions

**Comments to the Author**

1. Is the manuscript technically sound, and do the data support the conclusions?

Reviewer #1: Yes

Reviewer #2: Yes

2. Has the statistical analysis been performed appropriately and rigorously?

Reviewer #1: Yes

Reviewer #2: Yes

3. Have the authors made all data underlying the findings in their manuscript fully available?

Reviewer #1: Yes

Reviewer #2: Yes

4. Is the manuscript presented in an intelligible fashion and written in standard English?

Reviewer #1: Yes

Reviewer #2: Yes

Reviewer #1: This paper presents a valuable study evaluating the clinical and microstructural outcomes of vitrectomy for lamellar macular hole (LMH). The findings are significant as they highlight not only the improvement in visual acuity but also the time course of resolution of intraretinal cysts (Inner/Outer Cysts) and identify specific microstructural features—namely Epiretinal Proliferation (EP) and Ellipsoid Zone (EZ) disruption—as important prognostic factors. I believe this paper will be acceptable with a few more revisions.

Major points

To more clearly identify the true prognostic factors for visual outcome (e.g., final 6-month BCVA), please consider to perform univariate and multivariate regression analyses using the baseline (preoperative) characteristics (i.e. including BCVA, EZ status, EP status, and cyst status) as variables. It would be a great interest for the readers. If it is difficult to conduct in the study, please include the reasons for that in the limitation.

It is unclear whether the inclusion of data points from Month 1 to Month 6 in Table 2 is necessary or informative, given the study's primary objective of identifying prognostic factors. Please consider to remove these data if it is unnecessary for authors' discussion and conclusions.

Minor points

The Results section appears to discuss subtypes or specific forms of LMH, yet the Methods section only states that diagnosis was 'based on OCT.' The criteria for distinguishing and classifying these specific LMH subtypes must be clearly defined in the Methodology. Similarly, the changes in OCT parameters, particularly those related to EP, EZ disruption and cyst, constitute a highly significant finding of this manuscript. Although image examples are provided in the Methods section, detailed descriptions of how these features were defined are lacking and should be included.

It would be beneficial to include a more detailed description of the OCT imaging protocol. For instance, were these assessment based solely on a single B-scan slice through the macula, or were multiple slices or volume scans reviewed to determine the presence and extent of the pathology?

The Statistical Analysis section should be expanded to include greater detail. While the results indicate comparisons across multiple time points, it is unclear whether the authors employed multiple comparison adjustments (e.g., Bonferroni or Holm's correction) when conducting comparisons between different time points (preoperative vs. 1 month, 1 month vs. 3 months, etc.). Clarification on the specific post-hoc tests used is necessary to ensure the validity of the reported p-values.

"ERM-F0053zzz" should be typo in Table 1.

The sample size (n) for each group in Table 2 must be clearly indicated.

The 0.02 in the last line in Table 2 should be bold and with * because it is  < 0.05.

Reviewer #2: Goto et al. investigated changes in visual function and OCT parameters following vitrectomy for LMH. The Introduction is appropriate, and the Methods, Results, and Discussion are sufficiently described. The manuscript would become even stronger by addressing the following points:

1. Six eyes with glaucoma were included in the study. Please clarify whether these cases had no clinically significant visual field loss or optic nerve damage that could affect visual acuity. If eyes with advanced glaucoma were included, exclusion or subgroup analysis may be warranted, as glaucoma could confound postoperative visual outcomes.

2. In Table 2, it would be more informative to present the actual number of eyes with and without each OCT abnormality (e.g., inner/outer retinal cysts, EP, EZ disruption) at each time point. Providing case numbers in addition to percentages would enhance clarity and interpretability for readers.

3. Were there any differences in postoperative visual outcomes or OCT improvement between eyes that received air tamponade and those that did not? A brief subgroup comparison or statement confirming no meaningful difference would strengthen the clinical applicability of the findings.

4. Based on the present results, it may be helpful to discuss practical clinical indicators or decision-making points when considering surgery for LMH—such as preoperative BCVA, presence of EZ disruption or EP, progression of anatomical changes, or persistence of cystic alterations. Including such considerations in the Discussion would increase the manuscript’s clinical relevance.

**Do you want your identity to be public for this peer review?** For information about this choice, including consent withdrawal, please see our Privacy Policy

Reviewer #1: No

Reviewer #2: No

---

## [Author Response · Author response to Decision Letter 1]

22 Dec 2025

Responses to Reviewer 1 Comments

Question 1: To more clearly identify the true prognostic factors for visual outcome (e.g., final 6-month BCVA), please consider to perform univariate and multivariate regression analyses using the baseline (preoperative) characteristics (i.e. including BCVA, EZ status, EP status, and cyst status) as variables. It would be a great interest for the readers. If it is difficult to conduct in the study, please include the reasons for that in the limitation.

Response 1: We greatly appreciate the reviewer's insightful question. We performed a multiple linear regression analysis to identify factors associated with 6-month postoperative BCVA, including baseline OCT morphological features and preoperative BCVA. As a result, preoperative BCVA was the only factor significantly associated with 6-month postoperative BCVA (p = 0.0052). Due to limitations on the number of tables and figures, we have included these results in the Results section instead of presenting a separate table. Although both EZ disruption and EP were associated with poorer postoperative BCVA in Table 2, these associations disappeared after adjustment for preoperative BCVA and other structural factors in the multivariate model. This may suggest that the apparent effects of EZ disruption and EP reflect baseline retinal status rather than representing independent prognostic determinants. We have added this explanation to the Discussion part.

Change 1: We have added the following sentences in the manuscript.

Additionally, we performed a multiple linear regression analysis to identify factors associated with the 6-month postoperative BCVA, including preoperative BCVA and OCT parameters. The adjusted R² was 0.35, and the variance inflation factor for all variables was below 1.31, indicating no significant multicollinearity. Among the predictors, worse preoperative BCVA was significantly associated with worse postoperative BCVA (p = 0.0052). (Page 8, lines 165–169)

First, the postoperative follow-up period was relatively short (6 months), and some cases were lost to follow-up, which may have reduced the statistical power of subsequent analyses. Furthermore, although EZ disruption and EP were associated with poorer postoperative BCVA, these associations were not observed in the multivariate analysis after adjusting for preoperative BCVA. This may suggest that their apparent effects reflect baseline retinal status rather than independent prognostic determinants. A larger sample size will be necessary to clarify their true influence. (Page 13, lines 287–293)

Question 2: It is unclear whether the inclusion of data points from Month 1 to Month 6 in Table 2 is necessary or informative, given the study's primary objective of identifying prognostic factors. Please consider to remove these data if it is unnecessary for authors' discussion and conclusions.

Response 2: Thank you for your valuable comment. Giansanti F et al. reported that visual acuity continues to improve from 1 to 6 months after surgery and even later, indicating a slow and sustained retinal healing process【18】. These findings suggest that postoperative data from 1 to 6 months may offer additional prognostic insight. For this reason, we consider it appropriate to include these time-point data in Table 2.

Change 2: We have added the following sentences in the manuscript. Additionally, references [18] were included.

Giansanti F et al. reported that visual acuity continues to improve from 1 to 6 months after surgery and even later, indicating a slow and sustained retinal healing process【18】. (Page 10, lines 218–220)

Question 3: The Results section appears to discuss subtypes or specific forms of LMH, yet the Methods section only states that diagnosis was 'based on OCT.' The criteria for distinguishing and classifying these specific LMH subtypes must be clearly defined in the Methodology. Similarly, the changes in OCT parameters, particularly those related to EP, EZ disruption and cyst, constitute a highly significant finding of this manuscript. Although image examples are provided in the Methods section, detailed descriptions of how these features were defined are lacking and should be included.

Response 3: We appreciate the reviewer’s valuable comment. We have added descriptions of the diagnostic criteria for the LMH-related subtypes ERM, ERM-FS, and MPH to the Patient data and OCT analysis section of the Methods. In addition, we have defined EP, EZ disruption, and intraretinal cysts in accordance with previously established OCT-based criteria.

Change 3: We have revised the sentences as follows:

ERM was defined as hyperreflective proliferation on the surface of the ILM, and the associated conditions, such as MPH and ERM-FS, were also classified based on updated criteria proposed by the recent consensus [4]. Eyes meeting more than two diagnostic criteria for different associated diseases across OCT B-scan images were considered overlapping cases and included in the analysis of each corresponding group. (Page 4, lines 79–83)

Inner and outer retinal cysts were defined as hyporeflective, small, round or elliptical spaces located in the inner nuclear layer and outer nuclear layer, respectively. EP was defined as homogeneous isoreflective epiretinal material over the ILM, and EZ disruption was defined as discontinuity of the EZ line. (Page 4, lines 86–89)

Question 4: It would be beneficial to include a more detailed description of the OCT imaging protocol. For instance, were these assessment based solely on a single B-scan slice through the macula, or were multiple slices or volume scans reviewed to determine the presence and extent of the pathology?

Response 4: We appreciate the reviewer’s helpful comment. In response, we have added a more detailed description of the OCT imaging protocol in the Methods section, specifying that both horizontal and vertical B-scan images were obtained and evaluated.

We also acknowledge that our classification was based solely on B-scan images, which we have noted as a limitation in the Discussion, as this approach may reduce the precision of disease differentiation.

Change 4: We have revised the relevant portion and added the following sentences:

Both horizontal and vertical OCT B-scan images were obtained. (Page 4, line 76)

Question 5: The Statistical Analysis section should be expanded to include greater detail. While the results indicate comparisons across multiple time points, it is unclear whether the authors employed multiple comparison adjustments (e.g., Bonferroni or Holm's correction) when conducting comparisons between different time points (preoperative vs. 1 month, 1 month vs. 3 months, etc.). Clarification on the specific post-hoc tests used is necessary to ensure the validity of the reported p-values.

Response 5: We appreciate the reviewer’s helpful comment. To clarify, comparisons of OCT parameters across time points were performed using McNemar’s test for paired proportions. Based on the study objective, three specified comparisons were conducted (Pre vs 1M, Pre vs 3M, and Pre vs 6M). Bonferroni correction was applied to adjust for multiple testing (α = 0.05/3 = 0.0167). After correction, significant reductions in outer retinal cysts were observed at 3 months (p = 0.00328) and 6 months (p = 0.00937), whereas no significant change was found at 1 month (p = 0.114). These results indicate that outer retinal cysts do not change substantially at 1 month but show a significant decrease from 3 months onward. We have added this explanation to the Statistical Analysis section. In addition, Figure 3 has been revised accordingly.

Change 5: We have revised the sentences as follows:

For the longitudinal comparison of categorical OCT parameters, three pre-specified paired comparisons (Pre vs 1M, Pre vs 3M, and Pre vs 6M) were evaluated, and Bonferroni correction was applied (adjusted significance threshold: P < 0.0167). All other tests of association were considered statistically significant at P<0.05. The analyses were performed using SPSS Statistics version 29.0 (IBM Corp., Armonk, NY, USA). (Page 6, lines 127–131)

McNemar’s test with Bonferroni correction was applied. Significant differences were observed only for outer retinal cysts between preoperative and 3 months (p = 0.00328) and between preoperative and 6 months (p = 0.00937). * p < 0.0167 (Bonferroni-corrected). (Page 9, lines 185–188)

Question 6. "ERM-F0053zzz" should be typo in Table 1.

Response 6: Thank you for your valuable comment. We have revised the sentences.

Change 6: We have revised the sentences as follows:

+ ERM-FS (Page 7, line 146)

Question 7. The sample size (n) for each group in Table 2 must be clearly indicated.

Response 7: Thank you for your valuable comment. We have revised Table 2 to clearly indicate the sample size (n) for each group. During this revision, we also recognized that the dropout numbers up to 6 months postoperatively had not been explicitly stated, and we have now added the number of cases analyzed at each time point to the Results section. In addition, errors in the case counts for preoperative EP prevalence and EZ disruption were corrected. However, these corrections did not affect the statistical results.

Change 7: We have revised the sentences as follows and have updated Table 2 accordingly.

The number of eyes available for analysis at each follow-up time point was as follows: 42 eyes preoperatively, 40 at 1 month, 38 at 3 months, and 25 at 6 months. (Page 7, lines 140–142)

EP was observed preoperatively in 59.5% of the eyes. (Page 7, line 154)

The frequency of eyes with EZ disruption did not change significantly during the 6-month follow-up period. (Page 9, lines 184–185)

Question 8�The 0.02 in the last line in Table 2 should be bold and with * because it is  < 0.05.

Response 8: Thank you for your valuable comment. We have revised the Table 2.

Change 8: We have revised the Table 2 as follows:

< 0.05*(Page 9, line 191)

In addition, the Abstract was revised accordingly to reflect these changes.

Responses to Reviewer 2 Comments

Question 1: Six eyes with glaucoma were included in the study. Please clarify whether these cases had no clinically significant visual field loss or optic nerve damage that could affect visual acuity. If eyes with advanced glaucoma were included, exclusion or subgroup analysis may be warranted, as glaucoma could confound postoperative visual outcomes.

Response 1: Thank you for this important comment. All six eyes with glaucoma included in the study met the criteria for early-stage disease, with MD better than −3 dB and no visual field defects within the central 5 degrees on static perimetry. Eyes with advanced glaucoma that could affect central visual acuity were not included in the analysis. Therefore, the impact of glaucoma on the postoperative outcomes is expected to be minimal. Regarding this point, we have added a clarification in the Results section.

Change 1: We have revised the sentences as follows:

Six eyes with glaucoma were also included. All of these had early-stage disease, with MD values better than −3 dB and no visual field loss within the central 5 degrees. Eyes with more advanced glaucoma that could affect central visual acuity were not included in the analysis. (Page 6, lines 142–145)

Question 2: In Table 2, it would be more informative to present the actual number of eyes with and without each OCT abnormality (e.g., inner/outer retinal cysts, EP, EZ disruption) at each time point. Providing case numbers in addition to percentages would enhance clarity and interpretability for readers.

Response 2: Thank you for your valuable comment. We have revised Table 2 to clearly indicate the sample size (n) for each group. During this revision, we also recognized that the dropout numbers up to 6 months postoperatively had not been explicitly stated, and we have now added the number of cases analyzed at each time point to the Results section. In addition, errors in the case counts for preoperative EP prevalence and EZ disruption were corrected. However, these corrections did not affect the statistical results.

Change 2: We have revised the sentences as follows and have updated Table 2 accordingly.

The number of eyes available for analysis at each follow-up time point was as follows: 42 eyes preoperatively, 40 at 1 month, 38 at 3 months, and 25 at 6 months. (Page 7, lines 140–142)

EP was observed preoperatively in 59.5% of the eyes. (Page 7, line 154)

The frequency of eyes with EZ disruption did not change significantly during the 6-month follow-up period. (Page 9, lines 184–185)

Question 3: Were there any differences in postoperative visual outcomes or OCT improvement between eyes that received air tamponade and those that did not? A brief subgroup comparison or statement confirming no meaningful difference would strengthen the clinical applicability of the findings.

Response 3: Thank you for your valuable comment. We performed subgroup comparisons between eyes that received air tamponade and those that did not, evaluating postoperative BCVA and OCT findings at each time point. Due to limitations on the number of tables and figures, the results have been summarized in the Results section, and a supplementary table has been provided to present the detailed data.

Change 3: We have revised the sentences as follows, and have prepared a supplementary table.

In the subgroup analysis comparing eyes that received air tamponade (n = 30) with those that received BSS tamponade (n = 12), no significant differences in postoperative BCVA or OCT findings were observed at any follow-up time point. (Page 8, lines 162–164)

Question 4: Based on the present results, it may be helpful to discuss practical clinical indicators or decision-making points when considering surgery for LMH—such as preoperative BCVA, presence of EZ disruption or EP, progression of anatomical changes, or persistence of cystic alterations. Including such considerations in the Discussion would increase the manuscript’s clinical relevance.

Response 4: We appreciate the reviewer’s comment. We have added interpretations in the Discussion regarding preoperative BCVA, EZ disruption, EP, and cystic or anatomical changes, clarifying how these findings may serve as practical clinical indicators when considering surgery for LMH.

Change 4: We have revised the sentences as follows:

Based on these findings, it is essential to avoid missing the appropriate timing for surgery and to inform patients that postoperative visual improvement may require some time. In addition, because preoperative BCVA appears to be an important indicator of postoperative visual outcomes, it may serve as a useful reference when determining the timing of surgical intervention. (Page 11, lines 220–224)

In this study, cystic changes alone appeared to have limited value in guiding surgical decision-making for LMH and were better interpreted together with other structural findings. (Page 12, lines 255–257)

These findings suggest that eyes with preserved EZ integrity are more likely to benefit from surgery, whereas substantial EZ disruption may limit postoperative improvement, making EZ status a useful clinical indicator when considering surgical timing and expected outcomes. (Page 12, lines 267–270)

Specifically, a prior report [33] indicated that patients without EP achieve better postoperative visual outcomes than those with EP, which is consistent with the association observed in our study. However, this relationship was no longer evident after adjusting for preoperative BCVA, suggesting that EP may reflect baseline retinal status rather than serving as an independent prognostic factor. (Page 13, lines 274–278)

In addition, the Abstract was revised accordingly to reflect these changes.

---

## [Decision Letter · Decision Letter 1]

27 Jan 2026

Changes in visual acuity and retinal microstructures following vitrectomy for lamellar macular hole

PONE-D-25-46076R1

Dear Dr. Kubota,

We’re pleased to inform you that your manuscript has been judged scientifically suitable for publication and will be formally accepted for publication once it meets all outstanding technical requirements.

Kind regards,

Tatsuya Inoue

Academic Editor

PLOS One

Additional Editor Comments (optional):

Reviewers' comments:

Reviewer's Responses to Questions

**Comments to the Author**

Reviewer #1: All comments have been addressed

Reviewer #2: All comments have been addressed

2. Is the manuscript technically sound, and do the data support the conclusions?

Reviewer #1: Yes

Reviewer #2: Yes

3. Has the statistical analysis been performed appropriately and rigorously?

Reviewer #1: Yes

Reviewer #2: I Don't Know

4. Have the authors made all data underlying the findings in their manuscript fully available?

Reviewer #1: Yes

Reviewer #2: Yes

5. Is the manuscript presented in an intelligible fashion and written in standard English?

Reviewer #1: Yes

Reviewer #2: Yes

Reviewer #1: The authors have addressed all of my concerns and suggestions comprehensively. The revisions made have significantly improved the clarity and quality of the manuscript. I have no further comments and recommend the paper for publication.

Reviewer #2: The authors have adequately responded to my comments, and I believe the manuscript has improved significantly and is now suitable for publication.

**Do you want your identity to be public for this peer review?** For information about this choice, including consent withdrawal, please see our Privacy Policy

Reviewer #1: No

Reviewer #2: No

---

## [Editor Report · Acceptance letter]

PONE-D-25-46076R1

PLOS One

Dear Dr. Kubota,

I'm pleased to inform you that your manuscript has been deemed suitable for publication in PLOS One. Congratulations! Your manuscript is now being handed over to our production team.

Kind regards,

on behalf of

Dr. Tatsuya Inoue

Academic Editor

PLOS One